# Effects of Jerusalem Artichoke (*Helianthus tuberosus*) as a Prebiotic Supplement in the Diet of Red Tilapia (*Oreochromis* spp.)

**DOI:** 10.3390/ani12202882

**Published:** 2022-10-21

**Authors:** Clara Trullàs, Mariya Sewaka, Channarong Rodkhum, Nantarika Chansue, Surintorn Boonanuntanasarn, Manoj Tukaram Kamble, Nopadon Pirarat

**Affiliations:** 1Wildlife, Exotic and Aquatic Animal Pathology Research Unit, Department of Pathology, Faculty of Veterinary Science, Chulalongkorn University, Bangkok 10330, Thailand; 2Faculty of Veterinary Science, Rajamangala University of Technology Srivijaya, Nakhon Si Thammarat 80240, Thailand; 3Center of Excellence in Fish Diseases (CE FID), Department of Veterinary Microbiology, Faculty of Veterinary Science, Chulalongkorn University, Bangkok 10330, Thailand; 4Department of Veterinary Medicine, Faculty of Veterinary Science, Chulalongkorn University, Bangkok 10330, Thailand; 5Institute of Agricultural Technology, School of Animal Production Technology, Suranaree University of Technology, Nakhon Ratchasima 30000, Thailand

**Keywords:** Jerusalem artichoke, prebiotic, red tilapia, growth performance, intestinal morphology, antioxidant-related genes, *Aeromonas veronii*

## Abstract

**Simple Summary:**

Factors such as changes in water quality, the transport and handling of the fish, the presence of pollutants, and the high densities used in intensive farming of red tilapia, among others, can cause oxidative stress in fish, which can lead to immune system suppression and increase the risk of opportunistic infections. The use of Jerusalem artichoke (JA) as an oligofructose-rich prebiotic dietary supplement in fish feed could enhance growth performance, hematological and immunological parameters, as well as disease resistance. However, the effect of JA in diets for red tilapia is poorly reported. In the present study, we evaluated the effects of a Jerusalem artichoke-supplemented diet on the blood chemistry, growth performance, intestinal morphology, expression of antioxidant-related genes, and disease resistance against *Aeromonas veronii* challenge in juvenile red tilapia. The results of JA-supplemented (JA5 and JA10) diets showed beneficial effects in terms of growth performance, blood chemistry, and intestinal morphology. Importantly, the prebiotic diets (JA5 and JA10) were associated with a significant increase in the expression of *gpx1* and *gst* antioxidant-related genes and disease resistance against *A.*
*veronii*. In conclusion, prebiotic diets have the potential to be employed as prospective supplements in sustainable red tilapia farming.

**Abstract:**

The aim of the present study was to evaluate the effects of a Jerusalem artichoke-supplemented diet on the blood chemistry, growth performance, intestinal morphology, expression of antioxidant-related genes, and disease resistance against *Aeromonas veronii* challenge in juvenile red tilapia. A completely randomized design (CRD) was followed to feed red tilapias with three experimental diets: control, 5.0 g/kg JA-supplemented (JA5), or 10.0 g/kg JA-supplemented (JA10) diets in triplicates for 4 weeks. The results revealed that the growth performance, weight gain (WG), specific growth rate (SGR), and average daily gain (ADG) of fish fed diets JA5 and JA10 were significantly higher (*p* < 0.05) than those of fish fed the control diet. Fish fed the control diet had significantly higher T-bilirubin, D-bilirubin, and ALT in blood serum than fish fed JA5 and JA10, as well as higher BUN than fish fed JA5. The number of goblet cells in the proximal and distal parts of the intestine revealed that the number of acid, neutral, and double-staining mucous cells of fish fed diets JA5 and JA10 was significantly higher (*p* < 0.05) than in fish fed the control diet. The diets including the prebiotic (JA5 and JA10) were associated with a significant increase in the expression of *gpx1* and *gst* antioxidant-related genes and disease resistance against *A. veronii* in juvenile red tilapia. Therefore, JA5 and JA10 can be employed as promising prebiotics for sustainable red tilapia farming.

## 1. Introduction

Red tilapia is recognized as an important source of animal protein globally due to its fast growth, tolerance for poor water quality, and disease resistance [1,2]. The culture of this species has increased rapidly, and it is an economically important fish species worldwide. However, intensive red tilapia farming is associated with stressors such as poor water quality and hypoxia, which lead to oxidative stress in fish [3,4]. Thus, these factors play an important role in immune system suppression and increase the risk of opportunistic infections, which are the major cause of economic losses in red tilapia culture [5,6,7].

Prebiotics are naturally produced and environmentally friendly, and their administration has been cited as a potential method to prevent fish infections in the aquaculture industry [8,9,10]. Prebiotics are defined as non-digestible food ingredients that beneficially affect the host’s health by selectively stimulating the growth and/or activity of beneficial microflora in the colon. Food ingredients with prebiotic properties have unique characteristics such as limited hydrolysis and absorption in the upper gastrointestinal tract, as well as selective stimulation of beneficial intestinal bacteria [11]. Prebiotics are found in several foods such as Jerusalem artichokes (JA), leeks, asparagus, chicory, garlic, onions, wheat, oats, and soybeans [12].

The JA (*Helianthus tuberosus*) or Kantawan, a member of the family *Asteraceae*, is a root vegetable native to central-eastern North America [13] that contains inulin and fructooligosaccharides (FOS), two of the most common prebiotics [14,15]. Inulin is a carbohydrate with low digestibility that is highly used in feeds for livestock and aquatic animals [16,17,18]. The FOSs are also included in animal feeds and have been shown to significantly increase fecal bifidobacteria at relatively low levels of consumption [19]. As reported, the use of JA as a dietary supplement in fish feed could enhance growth performance, hematological, and immunological parameters, as well as disease resistance [19,20,21]. Moreover, JA tubers are composed of natural antioxidants, including phenolic compounds with potential free radical scavenging activity [22,23,24,25], which could improve immune function and balance the excessive production of reactive oxygen species during infections [26,27]. However, the effect of JA on diets for red tilapia is poorly reported. Therefore, the aim of the present study was to investigate the effect of JA-supplemented diets on the blood chemistry, growth performance, intestinal morphology, and expression of antioxidant-related genes in juvenile red tilapia.

## 2. Materials and Methods

### 2.1. Preparation of Experimental Diets

The JA samples were obtained from Phetchabun Research Station, Agro-Ecological System Research and Development Institute, Kasetsart University, Thailand. The JA tubers were cleaned and sliced into thin pieces. These slices were then ground into powder using a hammer grinder. The samples were dried at 50 °C for 24 h and kept at 4 °C until use. The chemical composition and oligosaccharide contents of the JA tubers, such as dry matter, crude protein, crude lipid, crude fiber, ash, and fructans, were 934.4, 57.8, 1.7, 126.0, and 80.8 g/kg (dry matter basis), respectively [20]. Three experimental diets were prepared as follows: a control diet (un-supplemented), a 5.0 g/kg JA-supplemented diet (JA5), and a 10.0 g/kg JA-supplemented diet (JA10). The experimental diets (JA5 and JA10) and the control diet were prepared by thoroughly mixing 1 mL of the Jerusalem artichoke (powder dissolved in distilled water) in diets JA5 and JA10 and 1 mL of distilled water with 1 g of feed in the case of the commercial diet (Hi-grade 9951 Tilapia feed, Charoen Pokphand Foods Public Company Limited (CPF PCL), Samut Sakhon, Thailand), air-dried, and stored at 4 °C until feeding. The proximate composition of the commercial feed was: protein (30%), fat (3%), ash (8%), and moisture (12%).

### 2.2. Experimental Design

Two hundred and seventy monosex red tilapias with an average initial body weight of 14.1 ± 0.53 g were obtained from a Good Aquaculture Practices certified farm (CPF PCL, Samut Sakhon, Thailand) and acclimatized for two weeks before being distributed into nine 1000-L tanks (30 fish per tank). The completely randomized design (CRD) was followed to feed red tilapias with three experimental diets for 4 weeks. Water temperature was maintained at 25–28 °C throughout the experimental period, dissolved oxygen concentration ranged from 5.24–5.98 mg/L and pH was within the range 7.48–8.16. Fish were hand-fed 5% of their body weight twice a day, and we observed that the fish had entirely consumed the provided quantity of feed. Every two days, 50% of the water in each tank was replaced. The experiment was done in triplicates. All the experimental protocols were approved by the ethics committee of Chulalongkorn University Animal Care and Use Committee (CUACUC; Approval No. 1731039).

### 2.3. Growth Performance

Fish were weighed at the start and end of the experimental period. The final weight, weight gain (WG), specific growth rate (SGR), and feed conversion ratio (FCR) were calculated according to the following standard equations [28]:WG (%) = 100 × (final mean body weight – initial mean body weight)/initial mean body weight(1)
SGR = [(ln (final body weight) – ln (initial body weight)/days] × 100(2)
FCR = feed intake (g)/weight gain (g)(3)
ADG = (% gain)/(number of days)(4)

### 2.4. Blood Chemistry Analysis

After 4 weeks of feeding, six fish from each tank were randomly selected for blood sample collection. Samples were taken from the caudal vein using a hypodermic syringe and were allowed to clot at 4 °C for at least 3 h. Then, they were centrifuged at 2,600 × *g* for 10 min at room temperature to obtain the serum samples. Serum was analyzed for glucose, triglyceride, cholesterol, total protein, albumin, blood urea nitrogen (BUN), total bilirubin (T-bilirubin), direct bilirubin (D-bilirubin), serum alanine transaminase (ALT), and serum aspartate aminotransferase (AST) using an automated chemistry analyzer (AU400, Olympus, Tokyo, Japan).

### 2.5. Measurement of Villous Height, Villous Width, Absorptive Area, and Mucous Cells

After 4 weeks of feeding, six fish from each tank were randomly selected and anesthetized with clove oil (0.04 mL/L). Three parts of the intestine, the proximal (between the pyloric part of the stomach and the spiral part of the intestines), middle (the spiral part of the intestines), and distal parts (from the spiral part to 2 cm before the anus) were collected. Samples were fixed in neutral buffered 10% formalin, processed, and embedded in paraffin. Tissue sections (5 µm thick) were cut with a microtome and fixed on slides. To measure the villous height, the 10 highest intact villi were selected per section, and their height was measured from the tip to the bottom. The average height was expressed as the mean villous height for each section. The absorptive area was calculated by using the following Equation (5) [29]:absorptive area = villous height × villous width(5)

The mucous cells in the intestine were counted and classified by means of three types of special staining: Alcian Blue (AB) (pH 2.5) for acid mucin, Periodic Acid-Schiff (PAS) for neutral mucin, and AB-PAS double-staining for mixed mucin. The mucous cells from each section were counted using a high-power field (400 × magnification) and calculated as follows: mucous cell numbers/high-power field.

### 2.6. Antioxidant Gene Expression

After 4 weeks of feeding, total RNA was extracted from the livers of six randomly selected fish from each tank using TRIzol^™^ Reagent (Invitrogen, Carlsbad, CA, USA). The RNA was treated with the Ambion^®^ DNA-free™ DNase treatment and removal reagents (Ambion, Foster, CA, USA). The total RNA quality was then checked using a Nanodrop^®^ Lite spectrophotometer (Thermo Fisher Scientific Inc., Waltham, MA, USA) at an absorbance ratio A260/280 and assessed in 1.5% agarose electrophoresis. The 0.5 µg of total RNA obtained were converted to first-strand cDNA using ReverTra Ace^®^ qPCR RT Master Mix with gDNA Remover (Toyobo, Osaka, Japan) following the manufacturer’s instructions. All synthesized cDNA samples were preserved at −20 °C for quantitative PCR analysis.

Quantitative RT-PCR (qRT-PCR) for analysis of gene expression was performed by a Rotor-Gen Q (Qiagen, Germantown, MD, USA). The antioxidant-related genes *gpx1, gst, gr, cat,* and *sod* were used in this assay. The beta-actin gene (β-actin), a housekeeping gene, was applied as a reference gene for standardization of relative transcription levels. The primers are shown in Table 1.

The reaction mix consisted of 10 µL of KAPA SYBR^®^ FAST qPCR Master Mix (2×) Kit (Kapa Biosystems, Wilmington, MA, USA), 0.5 µM of forward and reverse primer, and 2 µL of cDNA and deionized water up to 20 mL of the final volume. All samples were analyzed in triplicate. The qRT-PCR cycling parameters were as follows: 95 °C for 3 min, followed by 40 cycles at 95 °C for 3 s, annealing at 60 °C for 20 s, extension at 72 °C for 20 s for all genes. Melt curve analysis was conducted to validate the specificity of the PCR amplification. Each transcript was analyzed using pooled samples for each group. The relative expression of antioxidant-related genes in the liver of fish after being fed the experimental diets were calculated by the 2^−^^ΔΔCt^ method using the Equation (6) [30]:ΔΔCt = (C_t, target gene_ – C_t, β__-actin_) _prebiotic_ – (C_t, target gene_ – C_t, β__-actin_) _control_(6)

### 2.7. Challenge Test

After 4 weeks of prebiotic diet feeding, 30 fish from each group were intraperitoneally injected (100 µL) with *A. veronii* at a final concentration of 10^7^ CFU/fish. *A. veronii* was isolated from naturally diseased Nile tilapia (*Oreochromis niloticus*) in Nong Khai Province, Northeastern Thailand and confirmed by a biochemical test and PCR assay [31]. After the challenge, clinical signs of infection or mortalities were recorded for 15 days. The cumulative survival rates were calculated by using the following Equation (7) [29]:Cumulative survival (%) = [((total fish – dead fish) × 100)/total fish](7)

### 2.8. Statistical Analysis

Results were analyzed by one-way analysis of variance (ANOVA) using SPSS (version 22, Armonk, NY, USA). The significance of the differences between means was tested by Duncan’s multiple range test. Kaplan-Meier survivorship curves and a log-rank (Mantel-Cox) test were used to compare the groups based on the cumulative survival percentage. Differences were considered significant when *p* < 0.05.

## 3. Results

### 3.1. Growth Performance

Red tilapia fed prebiotic diets showed better growth performance than those fed the control diet (Table 2).

The final weight, WG, SGR, and ADG of fish fed diets JA5 and JA10 were significantly higher (*p* < 0.05) than those of fish fed the control diet (Table 2). Moreover, the WG, SGR, and ADG results were higher in fish fed JA5 than in those fed JA10. In addition, fish fed JA5 or JA10 diets had significantly lower FCR (*p* < 0.05) than fish fed the control diet.

### 3.2. Blood Collection and Serum Chemistry Analysis

Results showed that fish fed JA5 had significantly higher (*p* < 0.05) glucose values than the control group (Table 3). Fish fed the control diet had the highest BUN value, which was significantly higher (*p* < 0.05) than that of fish fed diet JA5. In addition, fish from the control group had the highest T-bilirubin, D-bilirubin, and ALT values (*p* < 0.05).

### 3.3. Measurement of Villous Height, Villous Width, Absorptive Area, and Mucous Cells

The villous height and width along the intestine of red tilapia fed the JA-supplemented diets were higher than those in fish fed the control diet, but this was not statistically significant (*p* > 0.05) (Table 4).

However, the absorptive area in the distal part of the intestine presented a significant difference (*p* < 0.05), being higher in fish fed the JA-supplemented diets than in those fed the control diet.

The number of goblet cells in the proximal (Figure 1) and distal intestines revealed that the numbers of acid, neutral, and double-staining mucous cells in fish fed the JA-supplemented diets were significantly higher (*p* < 0.05) than those in fish fed the control diet (Table 5). In the middle intestine, the number of neutral mucous cells in fish fed the JA10 diet and the number of double-stained mucous cells in fish fed JA5 were significantly higher (*p* < 0.05) than those in fish fed the control diet.

### 3.4. Antioxidant Gene Expression

The gene expression profiles of the antioxidant-related genes *gpx1, gst, gr, cat* and *sod* are presented in Figure 2.

The JA5 prebiotic diet caused a significant increase (*p* < 0.05) in the expression of the *gpx1* gene (2.69-fold and 1.36-fold) compared to the Control and JA10 diets, while the JA10 diet resulted in a significant increase in the expression of the *gst* (4.2-fold) and *gr* (3.17-fold) genes compared to the Control diet. The expression of *cat* in fish fed the JA10 prebiotic diet was significantly higher (*p* < 0.05) (1.96-fold) than that in fish from the control diet, while the expression of *sod* did not present significant differences between the groups (*p* > 0.05).

### 3.5. Challenge Test

The Kaplan-Meier analysis for the prebiotic diets found significant differences (X^2^(2) = 18.820, *p* < 0.01) compared to the control (Figure 3).

The cumulative survival rate was significantly higher in the JA5 and JA10 groups, at 92% and 88%, respectively, when compared with the control (44%). After challenge with *A. veronii*, red tilapia mortality on day 1 occurred at rates of 12%, 4%, and 12% in the control, JA5, and JA10 groups, respectively. Moreover, fish mortality in the control, JA5, and JA10 groups stopped on day 13 (56%), day 6 (8%), and day 1 (12%), respectively.

## 4. Discussion

The beneficial effects of JA on the growth performance of Nile tilapia during the fingerling and juvenile stages has previously been reported [19,20]. In the present study, the higher WG, SGR, ADG, and FCR obtained in fish fed diets JA5 and JA10 compared to those of fish fed the control diet resulted similar to the results reported for juvenile Nile tilapia fed JA-supplemented diets for 56 days [19] and Asian seabass (*Lates calcarifer*) fed JA-supplemented diets for 45 days [29]. In addition, supplementing the diet with JA for 82 days improved the performance parameters of Nile tilapia during the fingerling stage [20]. The beneficial effects of JA on growth performance may be the result of compounds such as inulin, FOS, carbohydrate, protein, Vitamin C, and minerals [32]. In fact, previous studies reported that inulin improved the growth performance of Nile tilapia [19,20,33] and rainbow trout (*Oncorhynchus mykiss*), while FOS-supplemented diets improved the growth performance of rainbow trout [34] and turbot (*Psetta maxima*) [35].

Hematological and biochemical analysis is an inexpensive and useful tool for evaluating the physiological status and health of fish [36]. However, regarding the effects of prebiotics on such parameters, published results are often contradictory. As reported by Guerreiro et al. [37], the specific mechanisms of action responsible for the potential benefits of prebiotics are often difficult to elucidate due to the multiple modes of action and synergies that may occur. In the present study, even though fish fed the JA5 diet had a higher blood glucose level than the rest, fish fed the prebiotic diets had the lowest T-bilirubin, D-bilirubin, ALT, and BUN values, which could indicate a liver and kidney protective effect exerted by the prebiotic [38,39,40]. Regarding glucose, to the best of our knowledge, only Guerreiro et al. [41,42] have reported results on the effect of prebiotics on glucose metabolism in sea bass and sea bream. Regardless of their elucidations, prebiotics did not improve glucose tolerance in these two studies, and plasma glucose levels did not differ from those of the control group. As these same authors reported, there is scarce information on the effects of prebiotics on the glucose metabolism of fish; prebiotic effects on lipid metabolism have been more studied [37]. In spite of the lack of information on the potential protective effect of prebiotics on liver and kidney function, Yarahmadi et al. [43] reported the restoration of AST and ALT in rainbow trout fed a diet that included a combination of β-glucan and mannan oligosaccharides (MOS) and then infected with *A. hydrophila*. This indicated that the prebiotics had a protective effect on the fish with respect to disease. As described by Xu et al. [44] in rodents, the fermentation products of prebiotics can increase the production of mucin and regulate the action of hepatic lipogenic enzymes. Thus, prebiotics can exert synergistic effects together with the stimulated beneficial bacteria in the treatment of liver injury. The lack of significant differences in the rest of the blood parameters among the experimental groups indicated that the prebiotic-supplemented diets had no major effects on the blood serum biochemical parameters. This was in accordance with results reported by other authors for dietary prebiotics such as MOS [45,46,47,48], inulin [49], and FOS [50] in different fish species.

In terms of intestinal morphology, prebiotics may increase the absorptive area of the gastrointestinal tract based on changes in the microvillus height, among others [51,52,53]. Higher absorptive surface areas and higher microvilli densities in the intestine result in an increase in nutrient absorption [54], which may result in increased weight gain and feed efficiency [55]. The JA-supplemented diets had beneficial effects on the absorptive area in the distal part of the intestine of red tilapias. Previous studies have demonstrated that feeding JA-supplemented diets to juvenile Nile tilapia over 8 weeks improved the villous height in the proximal and middle parts of intestine [19], which was also observed in red drum fed dietary prebiotics for 8 weeks [53]. The fact that in our study differences were not present in the proximal and middle intestine could be due to the shorter duration of the experimental feeding compared to the aforementioned study. Hence, a longer experimental feeding period would be desired in order to assess the intestinal morphology. 

Goblet cells, also known as mucous cells, are the first line of defense of the intestinal mucosa, as they produce and secrete mucin, which forms a layer that leads to the physical removal of attached pathogens or toxins [56]. Mucous cells can be acid or neutral depending on the type of mucin secreted [57,58]. Acid mucous cells protect against bacterial translocation [59], while neutral mucous cells are linked to the digestion process [60]. In our study, the higher amounts of acid, neutral, and double-staining mucous cells observed in the proximal and distal intestines of fish fed the prebiotic-supplemented diets was in accordance with previous results reported in Nile tilapia fingerlings fed a JA-supplemented diet [61]. The increase in the three different types of mucous cells in the presence of prebiotic indicated a reinforcement of the intestinal immune system.

Reactive oxygen species released by inflammatory cells can cause damage to cellular macromolecules such as DNA, lipids, and proteins, leading to cell and tissue damage. Oxidative stress can be induced by heat, water pollution, and UV light, and cells need to produce antioxidant enzymes to fight against it [26]. Several studies have shown that JA has bioactive molecules such as phenolic compounds that have radical scavenging activity and SOD-like activity [22,23,45]. Genes *gpx1, gst, gr, cat*, and *sod* are the common antioxidant-related genes in fish [61,62]. The significant increase in the expression of the *gpx1, gst*, and *gr* genes in fish fed dietary JA in the present study was in line with the upregulation in the expression of the *gst* gene in Nile tilapia fed a β-glucans-supplemented diet [63]. Overall, feeding tilapia with JA improved the expression of antioxidant-related genes, which might ultimately be related to the observed improvement in the growth performance and disease resistance against *A. veronii*.

## 5. Conclusions

In conclusion, the results of the present study revealed that JA-supplemented diets improved growth performance, the amount of intestinal mucous cells, the expression of antioxidant-related genes and the disease resistance against *A. veronii* in juvenile red tilapia. Differences in the effects between the two prebiotic diets were not significant in most cases. Therefore, Jerusalem artichoke can be included as a prebiotic in diets for red tilapia.

## Figures and Tables

**Figure 1 animals-12-02882-f001:**
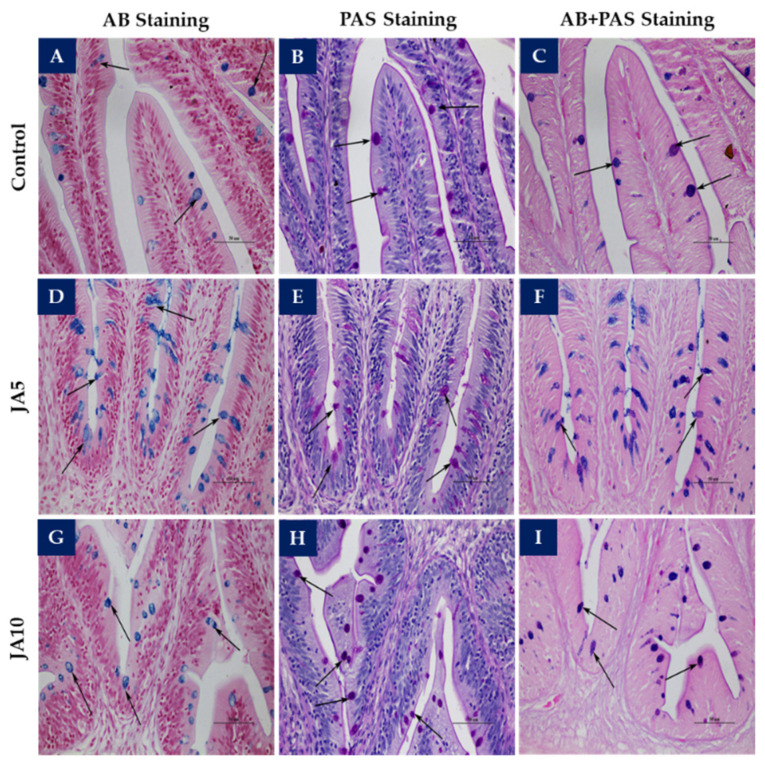
The proximal intestinal goblet cells (arrows) of red tilapia fed the control (**A**–**C**), JA5 (**D**–**F**), and JA10 diets (**G**–**I**) after special staining into three types: AB staining (**A**,**D**,**G**), PAS staining (**B**,**E**,**H**), and AB-PAS double-staining (**C**,**F**,**I**). Bar 50 µm. AB: Alcian Blue; PAS: Periodic acid-Schiff.

**Figure 2 animals-12-02882-f002:**
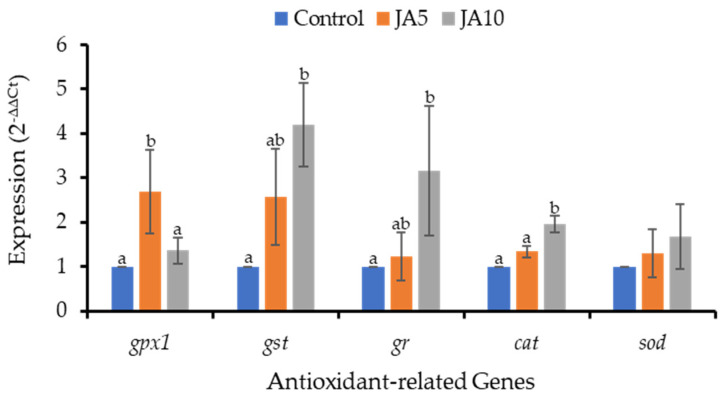
Relative expression of *gpx1, gst, gr, cat,* and *sod* in the liver of red tilapia fed experimental diets for 4 weeks. Results are means ± SD (*n* = 6). ^a, b^ Different superscript letters in each gene denote statistically significant differences between experimental diets (*p* < 0.05). JA5: 5.0 g/kg JA-supplemented diet; JA10: 10.0 g/kg JA-supplemented diet.

**Figure 3 animals-12-02882-f003:**
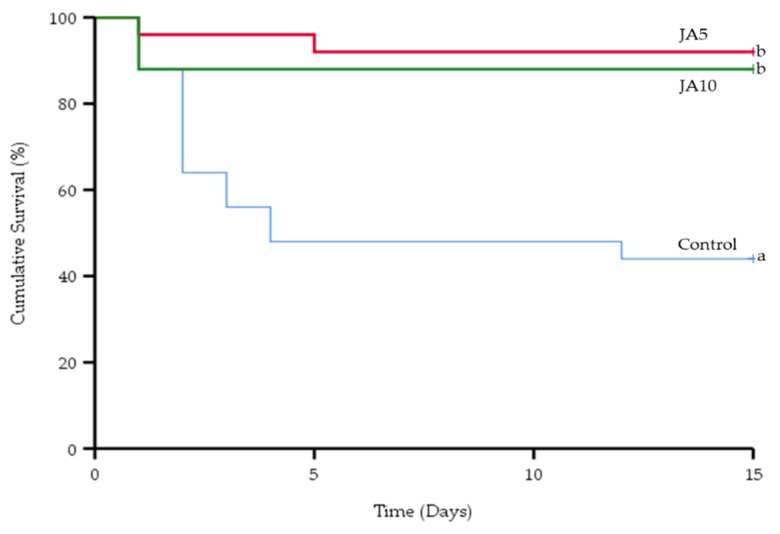
The average cumulative survival rate of red tilapia fed the control diet, JA5-supplemented diet, and JA10-supplemented diet for 4 weeks following *Aeromonas veronii* challenge. Kaplan-Meier survivorship curves over time (days) for red tilapia were plotted with the log-rank (Mantel-Cox) statistical test to compare the experimental groups. ^a, b^ Different superscript letters in each treatment denote statistically significant differences (*p* < 0.01). Data represent the means ± SD (*n* = 30).

**Table 1 animals-12-02882-t001:** Tilapia primers used for qRT-PCR in the current study.

Gene Name	qPCR Primer, Forward/Reverse (5′–3′)	Amplicon Size (bp)
*β* *-actin*	AAGGACCTGTACGCCAACAC	196
	ACATCTGCTGGAAGGTGGAC	
*gpx1*	GGAACGACAACCAGGGACTA	160
	TCCCTGGACGGACATACTTC	
*gst*	CAAAGGATCCCAAAGAACGA	184
	AGGTACACGGGTCCAGTCAG	
*gr*	GTTCCTCCCAGTAACGACCA	160
	TGTAGCAGTGCTCCCTTCCT	
*cat*	TGTTCCCATCCTTCATCCAT	182
	GAAGGTGTGAGAGCCGTAGC	
*sod*	AAAGGCATGTTGGAGACCTG	196
	AGACGTCCACCAGCATTACC	

**Table 2 animals-12-02882-t002:** Growth performance and feed utilization of red tilapia fed experimental diets for 4 weeks.

Parameters	Control	JA5	JA10
Initial weight (g)	14.68 ± 3.26	13.68 ± 5.15	13.88 ± 3.61
Final weight (g)	24.52 ± 5.67 ^a^	28.76 ± 5.37 ^b^	27.73 ± 6.18 ^b^
WG (%)	67.03 ± 7.0 ^a^	125.70 ± 45.46 ^c^	106.38 ± 13.79 ^b^
FCR	2.32 ± 0.55 ^c^	1.36 ± 0.11 ^a^	1.61 ± 0.35 ^b^
SGR (%/day)	1.76 ± 0.14 ^a^	2.74 ± 0.70 ^c^	2.49 ± 0.23 ^b^
ADG (% /day)	2.31 ± 0.24 ^a^	4.34 ± 1.57 ^c^	3.67 ± 0.48 ^b^

Results are means ± SD (*n* = 30); ^a, b, c^ Different superscript letters in a row denote statistically significant differences (*p* < 0.05). JA5: 5.0 g/kg JA-supplemented diet; JA10: 10.0 g/kg JA-supplemented diet; WG: weight gain; FCR: feed conversion ratio; SGR: specific growth rate; ADG: average daily gain.

**Table 3 animals-12-02882-t003:** Blood serum biochemical parameters of red tilapia fed experimental diets for 4 weeks.

Parameters	Control	JA5	JA10
Glucose (mg/dl)	45.17 ± 7.49 ^a^	75.67 ± 36.43 ^b^	50.67 ± 10.54 ^ab^
Triglycerides (mg/dl)	200.00 ± 97.51	187.67 ± 81.41	187.00 ± 48.53
Total cholesterol (mg/dl)	161.33 ± 10.54 ^a^	163.83 ± 29.63 ^a^	227.83 ± 24.60 ^b^
Total protein (g/dl)	2.60 ± 0.22	2.52 ± 0.44	2.83 ± 0.38
Albumin (g/dl)	0.97 ± 0.27	0.77 ± 0.12	0.97 ± 0.21
BUN (mg/dl)	1.80 ± 0.40 ^b^	1.00 ± 0.00 ^a^	1.50 ± 0.55 ^b^
Total bilirubin (mg/dl)	0.02 ± 0.01 ^b^	0.01 ± 0.00 ^a^	0.01 ± 0.00 ^a^
Direct bilirubin (mg/dl)	0.015 ± 0.001 ^b^	0.003 ± 0.001 ^a^	0.002 ± 0.00 ^a^
ALT (IU/l)	9.50 ± 6.53 ^b^	3.20 ± 1.17 ^a^	3.33 ± 0.82 ^a^
AST (IU/l)	13.00 ± 2.76	14.80 ± 5.38	17.33 ± 7.69

Results are means ± SD (*n* = 6)**.**
^a, b^ Different superscript letters in a row denote statistically significant differences (*p* < 0.05). JA5: 5.0 g/kg JA-supplemented diet; JA10: 10.0 g/kg JA-supplemented diet; BUN: blood urea nitrogen; ALT: alanine transferase; AST: aspartate aminotransferase.

**Table 4 animals-12-02882-t004:** Intestinal morphology of red tilapia fed the experimental diets for 4 weeks.

Parameters	Control	JA5	JA10
Proximal part			
Villi height (µm)	288.22 ± 6.96	293.82 ± 6.34	295.23 ± 5.98
Villi width (µm)	77.08 ± 3.20	76.71 ± 3.26	80.06 ± 1.45
Absorptive area (mm^2^)	0.0222 ± 0.001	0.0225 ± 0.0005	0.0235 ± 0.0002
Middle part			
Villi height (µm)	262.21 ± 2.72	263.92 ± 9.47	265.61 ± 8.39
Villi width (µm)	91.58 ± 7.42	94.29 ± 10.07	91.87 ± 5.00
Absorptive area (mm^2^)	0.0240 ± 0.0017	0.0248 ± 0.0017	0.0242 ± 0.0006
Distal part			
Villi height (µm)	172.96 ± 4.81	182.66 ± 3.62	178.89 ± 7.62
Villi width (µm)	89.20 ± 4.73	101.46 ± 4.66	104.14 ± 5.37
Absorptive area (mm^2^)	0.0154 ± 0.0006 ^a^	0.0185 ± 0.0005 ^b^	0.0185 ± 0.0009 ^b^

Results are means ± SD (*n* = 6). ^a, b,^ Different superscript letters in a row denote statistically significant differences (*p* < 0.05). JA5: 5.0 g/kg JA-supplemented diet; JA10: 10.0 g/kg JA-supplemented diet.

**Table 5 animals-12-02882-t005:** Average number of goblet cells for red tilapia fed experimental diets for 4 weeks.

Parameters	Control	JA5	JA10
Proximal part			
AB	27.84 ± 4.51 ^a^	46.00 ± 2.68 ^b^	47.50 ± 2.07 ^b^
PAS	30.75 ± 3.88 ^a^	58.37 ± 4.78 ^b^	61.83 ± 1.72 ^b^
AB-PAS double-staining	14.55 ± 1.37 ^a^	30.97 ± 2.23 ^b^	30.63 ± 0.95 ^b^
Middle part			
AB	57.00 ± 2.74 ^a^	56.80 ± 2.15 ^a^	56.80 ± 1.79 ^a^
PAS	56.60 ± 3.05 ^a^	57.40 ± 2.15 ^ab^	60.00 ± 1.58 ^b^
AB-PAS double-staining	48.00 ± 2.00 ^a^	51.00 ± 2.19 ^b^	49.50 ± 1.29 ^ab^
Distal part			
AB	49.00 ± 3.54 ^a^	88.50 ± 3.73 ^b^	89.23 ± 7.03 ^b^
PAS	56.44 ± 3.43 ^a^	100.17 ± 3.92 ^b^	100.33 ± 2.94 ^b^
AB-PAS double-staining	30.40 ± 1.67 ^a^	53.04 ± 1.93 ^b^	52.20 ± 3.56 ^b^

Results are means ± SD (*n* = 6). ^a, b^ Different superscript letters in a row denote statistically significant differences (*p* < 0.05). JA5: 5.0 g/kg JA-supplemented diet; JA10: 10.0 g/kg JA-supplemented diet.

## Data Availability

The data presented in this study are available from the corresponding author upon request.

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
