# Peer review of "Effects of Jerusalem Artichoke (Helianthus tuberosus) as a Prebiotic Supplement in the Diet of Red Tilapia (Oreochromis spp.)"

_animals, 2022, doi:10.3390/ani12202882_

Round 1

Reviewer 1 Report

Review comments

Title: Efficacy of Jerusalem Artichoke-Supplemented Diets on Blood 2 Chemistry, Growth Performance, Intestinal Morphology, Anti-3 oxidant-Related Genes Expression and Protection Against Aer-4 omonas veronii Challenge in Juvenile Red Tilapia (Oreochromis 5 spp.)

Journal: animals-1959001-peer-review-v1

+++++++++++++++++

Title to be change to “Dietary Jerusalem artichoke incorporation on growth Performance, blood parameters, intestinal morphology, anti-oxidant-related genes expression and disease resistsnce against Aeromonas veronii in red tilapia (Oreochromis spp.)”

Summary:

Line 25: “In this investigation” change to “In the present study”

Line 26, 34: “Jerusalem artichoke-supplemented” may be change to “Jerusalem artichoke-incorporated”

Line 28: “A. veronii” may be change to “Aeromonas veronii”

Line 31: “Aeromonas veronii” may be change to “A. veronii”

Abstract:

Line 48: “supplements” maybe change to ““supplement doses”

Keywords: “jerusalem artichoke” maybe change to “Jerusalem artichoke”

1. Introduction:

Line 62: Include citation “Devi et al., 2019a, b”

Devi G, Harikrishnan R, Paray BA, Al-Sadoon MK, Hoseinifar, S.H, Balasundaram C. Effect of symbiotic supplemented diet on innate-adaptive immune response, cytokine gene regulation and antioxidant property in Labeo rohita against Aeromonas hydrophila. Fish and Shellfish Immunology 2019a;89:687-700.

Line 68: Include citation “Ali et al., 2017

Ali SSR, Ambasankar K, Musthafa MS, Harikrishnan R. Jerusalem artichoke enriched diet on growth performance, immuno-hematological changes and disease resistance against Aeromonas hydrophila in Asian seabass (Lates calcarifer). Fish and Shellfish Immunology 2017;70:335-342

Line 73: “Fructooligosaccharides” maybe change to “FOSs”

Line 77, 80: Include citation “Ali et al., 2017

Line 81: “JA-supplemented” maybe change to “JA-incorporated”

Line 91: “a control diet”, the concentration or abbreviation not mention here…

Materials and methods:

Line 103: How many time was provide the feeds

Line 103: water replacement and unfed fecal matter not mention here..

“FCR= feed intake (g)/weight gain” maybe change to “FCR= feed intake (g)/weight gain (g)

Line 112: “puncture” include after caudal vein

Line 121: The concentration of “clove oil” is include here

Line 150: Describe reaction mix (10μl)

Line 161: How much volume injected in each fish; Describe subculture and purification, etc.

Discussion:

Line 251: Include “and Asian seabass fed JA- supplemented diets for 45 days (Ali et al., 2017)” after “for 56 days [17]”

Line 265: Include the mechanisms of action JA on fish hematology..

Author Response

Response to Reviewer 1 Comments

Title: Efficacy of Jerusalem Artichoke-Supplemented Diets on Blood 2 Chemistry, Growth Performance, Intestinal Morphology, Anti-3 oxidant-Related Genes Expression and Protection Against Aer-omonas veronii Challenge in Juvenile Red Tilapia (Oreochromis 5 spp.)

Journal: animals-1959001-peer-review-v1

+++++++++++++++++

Point 1: Title to be change to “Dietary Jerusalem artichoke incorporation on growth Performance, blood parameters, intestinal morphology, anti-oxidant-related genes expression and disease resistsnce against Aeromonas veronii in red tilapia (Oreochromis spp.)”

Response: Thank you for the valuable comment. We have improved the title after considering reviewer suggestions. The new title is “Effects of Jerusalem Artichoke (Helianthus tuberosus) as a Prebiotic Supplement in the Diet of Red Tilapia (Oreochromis spp.)”.

Summary:

Point 2: Line 25: “In this investigation” change to “In the present study”

Response: Thank you for the suggestion. We have made the suggested change in Line 24.

Point 3: Line 26, 34: “Jerusalem artichoke-supplemented” may be change to “Jerusalem artichoke-incorporated”

Response: Thank you for the suggestion. However, we believe that ‘supplemented’ is a better option than ‘incorporated’, as it illustrates the way of inclusion of the Jerusalem artichoke in the diets in a more precise way.

Point 4: Line 28: “A. veronii” may be change to “Aeromonas veronii”

Response: Thank you for the suggestion. We have made the suggested change in Line 26.

Point 5: Line 31: “Aeromonas veronii” may be change to “A. veronii”

Response: Thank you for the suggestion. We have made suggested change in Line 30.

Abstract:

Point 6: Line 48: “supplements” maybe change to ““supplement doses”

Response: Thank you for the suggestion. After a careful revision of the statement, we have expressed it as ‘prebiotics’, without using the word ‘supplements’, as we consider it is not needed (Line 46)

Point 7: Keywords: “jerusalem artichoke” maybe change to “Jerusalem artichoke”

Response: Thank you for the suggestion. We have made the suggested change in Line 47.

  1. Introduction:

Point 8: Line 62: Include citation “Devi et al., 2019a, b; Harikrishnan et al., 2021, 2022; Issac et al., 2021;”

Devi G, Harikrishnan R, Paray BA, Al-Sadoon MK, Hoseinifar, S.H, Balasundaram C. Effect of symbiotic supplemented diet on innate-adaptive immune response, cytokine gene regulation and antioxidant property in Labeo rohita against Aeromonas hydrophila. Fish and Shellfish Immunology 2019a;89:687-700.

Response: Thank you for the suggestion. According to the recommendation of the editorial office, we are not allowed to cite more than two references recommended by reviewers. Therefore, as per the suggestion of the reviewer, we have included Harikrishnan et al., 2022 [9] and Issac et al., 2021 [10] in Line 60.

Point 9: Line 68: Include citation “Ali et al., 2017”

Ali SSR, Ambasankar K, Musthafa MS, Harikrishnan R. Jerusalem artichoke enriched diet on growth performance, immuno-hematological changes and disease resistance against Aeromonas hydrophila in Asian seabass (Lates calcarifer). Fish and Shellfish Immunology 2017;70:335-342

Response: Thank you for the suggestion. According to the recommendation of the editorial office, we are not allowed to cite more than two references recommended by reviewers. Therefore, we cannot add the suggested reference from the reviewer.

Point 10: Line 73: “Fructooligosaccharides” maybe change to “FOSs”

Response: Thank you for the suggestion. However, abbreviations at the beginning of a sentence are to be avoided in scientific writing.

Point 11: Line 77, 80: Include citation “Ali et al., 2017”

Response: Thank you for the suggestion. According to the recommendation of the editorial office, we are not allowed to cite more than two references recommended by reviewers. Therefore, we cannot add the suggested reference from the reviewer.

Point 12: Line 81: “JA-supplemented” maybe change to “JA-incorporated”

Response: Thank you for the suggestion. However, we believe that ‘supplemented’ is a better option than ‘incorporated’, as it illustrates the way of inclusion of the Jerusalem artichoke in the diets in a more precise way.

Point 13: Line 91: “a control diet”, the concentration or abbreviation not mention here…

Response: Thank you for the comment. We did not include an abbreviation here because an abbreviation for the Control diet was not used in this manuscript. The concentration of the control diet was 0 g kg-1 i.e., un-supplemented. The commercial feed (Hi-grade 9951 Tilapia feed, CPF PCL, Thailand) was used as a control diet.

Line 91: We have included “control diet (un-supplemented)” in the revised manuscript.

Materials and methods:

Point 14: Line 103: How many time was provide the feeds

Response: The feeding frequency is stated in Line 107 (‘twice a day’).

Point 15: Line 103: water replacement and unfed fecal matter not mention here.

Response: Thank you for the comment. Every two days, 50% of the water in each tank was replaced. We observed that the fish had entirely consumed the provided quantity of feed; therefore, unfed fecal matter was not mentioned.

Lines 107-108: we have included “and observed that fish had entirely consumed the provided quantity of feed. Every two days, 50% of the water in each tank was replaced” in the revised manuscript. 

Point 16: “FCR= feed intake (g)/weight gain” maybe change to “FCR= feed intake (g)/weight gain (g)

Response: Thank you for the suggestion. We have made the changes in formula 3.

Point 17: Line 112: “puncture” include after caudal vein

Response: Thank you for the suggestion. However, we believe ‘puncture’ is not needed here, so we have left the sentence as it was.

Point 18: Line 121: The concentration of “clove oil” is include here

Response: Thank you for the suggestion. The concentration of clove oil (0.04 mL/L) has been added in Line 127.

Point 19: Line 150: Describe reaction mix (10μl)

Response: Thank you for the comment.

Lines 157-158: The reaction mix is described in the manuscript (‘The reaction mix consisted of 10 µL of KAPA SYBR® FAST qPCR Master Mix (2×) Kit (Kapa Biosystems, Massacusetts, USA), 0.5 µM of forward and reverse primer, and 2 µL of cDNA and deionized water up to 20 mL of the final volume’).

Point 20: Line 161: How much volume injected in each fish; Describe subculture and purification, etc.

Response: Thank you for the comment. A volume of 100 µL of A. veronii were intraperitoneally injected at a final concentration of 107 CFU/fish. A. veronii was isolated from naturally diseased Nile tilapia (Oreochromis niloticus) in Nong Khai Province, Northeastern Thailand and confirmed by biochemical test and PCR assay (Dong et al., 2015).

Lines 168-170: we have added “injected (100 µL) with A. veronii at a final concentration of 107 CFU/fish. Aeromonas. veronii was isolated from naturally diseased Nile tilapia (Oreochromis niloticus) in Nong Khai Province, Northeastern Thailand and confirmed by biochemical test and PCR assay [31]” in the revised manuscript.

Reference

  1. 31. Dong, H.T., Nguyen, V.V., Le, H.D., Sangsuriya, P., Jitrakorn, S., Saksmerprome, V., Senapin, S., Rodkhum, C. Naturally concurrent infections of bacterial and viral pathogens in disease outbreaks in cultured Nile tilapia (Oreochromis niloticus) farms. Aquaculture 2015, 448, 427-435.

Discussion:

Point 21: Line 251: Include “and Asian seabass fed JA- supplemented diets for 45 days (Ali et al., 2017)” after “for 56 days [17]”

Response: Thank you for the suggestion. The suggested changes “and Asian seabass (Lates calcarifer) fed JA- supplemented diets for 45 days [29]’ have been made in Lines 259-260.

Point 22: Line 265: Include the mechanisms of action JA on fish hematology.

Response: Thank you for the suggestion. Additional information on the effects of prebiotics on hematological and biochemical parameters has been added in the revised manuscript.

Lines 267-272: The hematological and biochemical analysis is an inexpensive and useful tool for evaluating the physiological status and health of fish [36]. However, regarding the effects of prebiotics on such parameter’s, published results are often contradictory. As reported by Guerreiro et al. [37], the specific mechanisms of action responsible for the potential benefits of prebiotics are often difficult to elucidate, due to multiple modes of action and synergies that may occur.

Lines 275-289: Regarding glucose, to the best of our knowledge, only Guerreiro et al. [41,42] reported results on the effect of prebiotics on the glucose metabolism in sea bass and sea bream, respectively. Regardless of their elucidations, prebiotics did not improve glucose tolerance in these two studies, and plasma glucose levels did not differ from those of the control group. As these same authors reported, there is scarce information on the effects of prebiotics on the glucose metabolism of fish, being prebiotic effects on lipid metabolism more studied [37]. In spite of the lack of information on the potential protective effect of prebiotics on the liver and kidney’s function, Yarahmadi et al. [43] reported the restoration of AST and ALT in rainbow trout fed a diet including a combination of β-glucan and MOS and then infected with A. hydrophila. This indicated a protective effect of the prebiotics on the fish in the case of disease. As described by Xu et al. [44] in rodents, the fermentation products of prebiotics can increase the production of mucin and regulate the action of hepatic lipogenic enzymes. Thus, the prebiotics can exert synergistic effects together with the stimulated beneficial bacteria in the treatment of liver injury.

Lines 291-293: This was in accordance with results reported by other authors for dietary prebiotics such as mannan oligosaccharides (MOS) [45–48], inulin [49] and fructooligosaccharides (FOS) [50] in different fish species.

References

  1. Witeska, M.; Kondera, E.; Ługowska, K.; Bojarski, B., Hematological methods in fish – Not only for beginners. Aquaculture 2022, 547, 737498.
  2. Guerreiro, I.; Oliva-Teles, A; Enes, P., Prebiotics as functional ingredients: focus on Mediterranean fish aquaculture. Reviews in Aquaculture 2017a, 0, 1-33.
  3. Guerreiro, I; Oliva-Teles, A; Enes, P; Improved glucose and lipid metabolism in European sea bass (Dicentrarchus labrax) fed short-chain fructooligosaccharides and xylooligosaccharides. Aquaculture 2015, 441, 57-63.
  4. Guerreiro, I; Serra, C.R.; Pousão-Ferreira, P; Oliva-Teles, A; Enes, P., Prebiotics effect on growth performance, hepatic intermediary metabolism, gut microbiota and digestive enzymes of white sea bream (Diplodus sargus). Aquaculture Nutrition 2017b, 24, 153-163.
  5. Yarahmadi, P.; Farsani, H.G.; Khazaei, A.; Khodadadi, M.; Rashidiyan, G.; Jalali, M.A., Protective effects of the prebiotic on the immunological indicators of rainbow trout (Oncorhynchus mykiss) infected with Aeromonas hydrophila. Fish and Shellfish Immunology 2016, 54, 589-597.
  6. Xu, S.; Zhao, M.; Wang, Q.; Xu, Z.; Pan, B.; Xue, Y.; Dai, Z.; Wang, S.; Xue, Z.; Wang, F.; Wang, F.; Xu, C., Effectiveness of probioitcs and prebiotics against acute liver injury: A Meta-Analysis. Frontiers in Medicine 2021, 8, 739337.
  7. Welker, T. L.; Lim, C.; Yildirim-Aksoy, M.; Shelby, R.; Klesius, P.H., Immune response and resistance to stress 694 and Edwardsiella ictaluri challenge in channel catfish, Ictalurus punctatus, fed diets containing commercial whole‐cell yeast or yeast subcomponents. Journal of the World Aquaculture Society 2007, 38 (1), 24-35.
  8. Sado, R.Y.; Bicudo, A.J.D.A.; Cyrino, J.P.E., Feeding dietary mannan oligosaccharides to juvenile Nile tilapia, Oreochromis niloticus, has no effect on hematological parameters and showed decreased feed consumption. Journal of the World Aquaculture Society 2008, 39, 821-826.
  9. Gultepe, N.; Hisar, O.; Salnur, S.; Hossu, B.; Tansel Tanrikul, T.; Aydm, S., Preliminary assessment of dietary mannanoligosaccharides on growth performance and health status of gilthead sea bream Sparus auratus. Journal of Aquatic Animal Health 2012, 24, 37-42.
  10. Razeghi Mansour, M.; Akrami, R.; Ghobadi, S.H.; Amani Denji, K.; Ezatrahimi, N.; Gharaei, A., Effect of dietary mannan oligosaccharide on growth performance, survival, body composition and some hematological parameters in giant sturgeon juvenile (Huso huso Linnaeus, 1754). Fish Physiology and Biochemistry 2012, 38, 829-835.
  11. Reza, A.; Abdolmajid, H.; Abbas, M.; Abdolmohammad, A.K., Effect of dietary prebiotic inulin on growth performance, intestinal microflora, body composition and hematological parameters of juvenile beluga Huso huso (Linnaeus, 1758). Journal of the World Aquaculture Society 2009, 40, 771-779.
  12. Hoseinifar, S.H.; Mirvaghefi, D.L.; Merrifield, B.M.; Amiri, S.; Yelghi, S.; Bastami, K.D., The study of some haematological and serum biochemical parameters of juvenile beluga (Huso huso) fed oligofructose. Fish Physiology and Biochemistry 2011, 37, 91-96.

Reviewer 2 Report

PFA

Author Response

Response to Reviewer 2 Comments

The MS captioned "Efficacy of Jerusalem Artichoke-Supplemented Diets on Blood Chemistry, Growth Performance, Intestinal Morphology, Anti- Oxidant-Related Genes Expression and Protection Against Aeromonas veronii Challenge in Juvenile Red Tilapia (Oreochromis spp.)" has been taken up by Trullas et al. It’s interesting that author has tried to corelate the things with multiple parameters however there are some of the shortfalls which are as follows.

Summary:

Point 1: line 21-22: It is not only the reason so sentence may be rewritten.

Response: Thank you for the suggestion. We have made changes in Lines 18-19 ”Factors such as changes in the water quality, transport and handling of the fish, presence of pollutants or the high densities used in intensive farming of red tilapia, among others”  

Confirm whether it is prebiotic or probiotic as it is not matching with scope of prebiotic. And at inclusion level of crude plant material what could be the % of a particular prebiotic as reposed.

Response: Jerusalem artichoke is a prebiotic, as it is rich in oligofructose. Oligofructose is an inulin-type fructan, which has been used as a prebiotic in aquaculture in many studies (Mahious et al., 2005, 2006; Ringø et al., 2006; Hoseinifar et al., 2011; Tiengtam et al., 2015).

A brief clarification “an oligofructose-rich” has been included in Line 21.

JA powder contained 502 g/kg fructan. The JA5 and JA10 diets were prepared to incorporate JA at 5.0 and 10.0 g/kg, respectively, which were equal to inulin levels of 2.5 and 5.0 g/kg, respectively (Boonanuntanasarn et al., 2018).

References

Boonanuntanasarn, S., Tiengtam, N., Pitaksong, T., Piromyou, P., & Teaumroong, N. (2018). Effects of dietary inulin and Jerusalem artichoke (Helianthus tuberosus) on intestinal microbiota community and morphology of Nile tilapia (Oreochromis niloticus) fingerlings. Aquaculture Nutrition24(2), 712-722.

Hoseinifar, S. H., Mirvaghefi, A., Mojazi Amiri, B., Rostami, H. K., & Merrifield, D. L. (2011). The effects of oligofructose on growth performance, survival and autochthonous intestinal microbiota of beluga (Huso huso) juveniles. Aquaculture Nutrition17(5), 498-504.

Mahious, A.S.; Ollevier, F., Probiotics and prebiotics in aquaculture: a review. The 1st Regional Workshop on Techniques for Enrichment of Live Food for Use in Larviculture 2005, 7–11 March, Urima, Iran, 17–26.

Mahious, A. S., Gatesoupe, F. J., Hervi, M., Metailler, R., & Ollevier, F. (2006). Effect of dietary inulin and oligosaccharides as prebiotics for weaning turbot, Psetta maxima (Linnaeus, C. 1758). Aquaculture International14(3), 219-229.

Ringø, E., Sperstad, S., Myklebust, R., Mayhew, T. M., & Olsen, R. E. (2006). The effect of dietary inulin on aerobic bacteria associated with hindgut of Arctic charr (Salvelinus alpinus L.). Aquaculture Research37(9), 891-897.

Tiengtam, N., Khempaka, S., Paengkoum, P., & Boonanuntanasarn, S. (2015). Effects of inulin and Jerusalem artichoke (Helianthus tuberosus) as prebiotic ingredients in the diet of juvenile Nile tilapia (Oreochromis niloticus). Animal Feed Science and Technology207, 120-129.

Point 2: Abstract: line 45-46 What do you mean by prebiotic diet. Is not clear why and how it has ben prepared?

Response: Thank you for the comment. The expression ‘prebiotic diet’ has been replaced by ‘prebiotic-supplemented diet’, as we agree with the fact that ‘prebiotic diet’ was not the correct term to use. The experimental diets (JA5 and JA10) and the control diet were prepared by thoroughly mixing 1 mL of the Jerusalem artichoke (powder dissolved in distilled water) in diets JA5 and JA10 and 1 mL of distilled water with 1 g of feed in the case of the commercial diet (Hi-grade 9951 Tilapia feed, CPF PCL, Thailand), air-dried and stored at 4° C until feeding (added in lines 92-97). 

Point 3: Line :47-48 it needs refinement as if you want to see the impact of prebiotics you might have studied the gut metagenomics for a clear-cut picture, however, its ok that you are relating it with the intestinal histology. If you have that data can be put here. It will aid value to the MS to great extent.

Response: Thank you for the observation. We have not studied gut metagenomics.

  1. Materials and Methods

Point 4: Author should have given the proximate composition of dietary component of plant material used and experimental diets.

Response: Thank you for your comment. The chemical composition and oligosaccharide contents of JA tuber, such as dry matter, crude protein, crude lipid, crude fiber, ash, and fructans, were 934.4, 57.8, 1.7, 126.0, and 80.8 g kg−1 (dry matter basis), respectively (Tiengtam et al., 2015) and has been added in Lines 88-90.

The experimental diets had the following proximate composition: protein (30%), fat (3%), ash (8%), and moisture (12%); and has been added in the Lines 96-97.

Reference

Tiengtam, N.; Khempaka, S.; Paengkoum, P.; Boonanuntanasarn, S., Effects of inulin and Jerusalem artichoke (Helianthus tuberosus) as prebiotic ingredients in the diet of juvenile Nile tilapia (Oreochromis niloticus). Animal Feed Science and Technology 2015, 207, 120–129.

Point 5: Why author used two experimental diets and statistically how it can be defined/ justified and how it would be CRD design explain.

Response: Thank you for the excellent observation. Previous studies reported that JA at 5.0 and 10.0 g/kg had beneficial effects on the growth and health of Nile tilapia juveniles (Tiengtam et al., 2015) and fingerlings (Tiengtam et al., 2017), and modulated the intestinal microbiota and morphology of Nile tilapia fingerlings (Boonanuntanasarn et al., 2018). Furthermore, the effect of JA at 5.0 and 10.0 g/kg on diets for red tilapia is poorly reported. Hence, we decided to use two experimental diets to investigate the effect of JA-supplemented diets on juvenile red tilapia.

A completely randomized design (CRD) is the basic experimental design that is used to study the effects of one factor, i.e. treatment or a fixed factor, while keeping others constant; therefore, it is often called a single-factor experiment (Bhujel, 2009). The present study experiment was conducted with three treatments: control (0g/kg), JA5 (5g/kg) and JA10 (10g/kg) in triplicates. Therefore, CRD was employed with three replicates (Table 2).

Table 2: Complete randomization of control, JA5 and JA10 of the experimental units.

Tank 1

C_R1

Tank 2

JA10_R1

Tank 3

JA5_R2

Tank 4

JA5_R3

Tank 5

C_R2

Tank 6

JA10_R1

Tank 7

JA10_R2

Tank 8

JA5_R1

Tank 9

C_R3

References

Bhujel, R. C. (2009). Statistics for aquaculture. John Wiley & Sons.

Boonanuntanasarn, S.; Tiengtam, N.; Pitaksong, T.; Piromyou, P.; Teaumroong, N., Effects of dietary inulin and Jerusalem artichoke (Helianthus tuberosus) on intestinal microbiota community and morphology of Nile tilapia (Oreochromis niloticus) fingerlings. Aquaculture Nutrition 2018, 24, (2), 712-722.

Tiengtam, N.; Khempaka, S.; Paengkoum, P.; Boonanuntanasarn, S., Effects of inulin and Jerusalem artichoke (Helianthus tuberosus) as prebiotic ingredients in the diet of juvenile Nile tilapia (Oreochromis niloticus). Animal Feed Science and Technology 2015, 207, 120–129.

Tiengtam, N.; Paengkoum, P.; Sirivoharn, S.; Phonsiri, K.; Boonanuntanasarn, S., The effects of dietary inulin and Jerusalem artichoke (Helianthus tuberosus) tuber on the growth performance, haematological, blood chemical and immune parameters of Nile tilapia (Oreochromis niloticus) fingerlings. Aquaculture Research 2017, 48, (10), 5280-5288.

Point 6: Is it ok to conduct the trial for four weeks to evaluate the impact of dietary Jerusalem Artichoke-Supplementation.

Response: Thank you for your comment. We consider that 4 weeks is the minimum feeding period used to evaluate the effect of a dietary additive in juvenile fish, as the potential effects of the additive on the animals will be already observed (Ardó et al., 2008; Mora-Sánchez et al., 2020; Yin et al., 2006). The juvenile red tilapia doubled their body weight during this period, which is an indicator of the adequacy of the trial duration. We agree, however, that the same trial performed in adult fish would require a longer period.

References

Ardó, L., Yin, G., Xu, P., Váradi, L., Szigeti, G., Jeney, Z., & Jeney, G. (2008). Chinese herbs (Astragalus membranaceus and Lonicera japonica) and boron enhance the non-specific immune response of Nile tilapia (Oreochromis niloticus) and resistance against Aeromonas hydrophilaAquaculture275(1-4), 26-33.

Mora-Sánchez, B., Fuertes, H., Balcázar, J. L., & Pérez-Sánchez, T. (2020). Effect of a multi-citrus extract-based feed additive on the survival of rainbow trout (Oncorhynchus mykiss) following challenge with Lactococcus garvieaeActa Veterinaria Scandinavica62(1), 1-4.

Yin, G., Jeney, G., Racz, T., Xu, P., Jun, X., & Jeney, Z. (2006). Effect of two Chinese herbs (Astragalus radix and Scutellaria radix) on non-specific immune response of tilapia, Oreochromis niloticusAquaculture253(1-4), 39-47.

Point 7:  What is prevalence of incidence of Aeromonas veronii infection in mentioned fish species.

Response: Thank you for your comment. Dong et al. (2015) reported that the A. veronii isolates were highly pathogenic to tilapia fingerlings and also suggested that it is an emerging pathogen affecting cultured tilapia farms in Thailand.

Reference

Dong, H. T., Nguyen, V. V., Le, H. D., Sangsuriya, P., Jitrakorn, S., Saksmerprome, V., ... & Rodkhum, C. (2015). Naturally concurrent infections of bacterial and viral pathogens in disease outbreaks in cultured Nile tilapia (Oreochromis niloticus) farms. Aquaculture448, 427-435.

Point 8: Whenever such nutritional related studies are conducted it is highly mandatory to put initial data of fish species used, its mention worthy.

Response: Thank you for your comment. We believed that all the initial data are present in the manuscript.

Lines 100-101: Two hundred and seventy monosex red tilapias with an average initial body weight of 14.1±0.53 g

Point 9: 3. Results Table 3, seems to be incomplete and I am not convinced wit the trend of data. If simply I ask the correlation or what is your inference out of this table ? As shown first of all u can find out two more parameters with these data. i.e. globulin and albumin/globulin ratio. As it is shown in table 2 the growth in terms of weight gain % was higher in JA5 but you total protein and albumin is significantly less in JA5 as compared with the JA10. Need to be explained and I suggest to recheck the values and to have a logical correlation in table 2 & 3. As it is known that globulin is an indication of immunity so after taking the value of globulin it also can be corelated with other immune parametrs.

*****One important observation is that in title it was mentioned that author did the challenge study but there is no reflection of it in nay table or figure it is very important as per the title.

Response: Thank you for the comment. Our purpose when including the blood serum biochemical parameters of the fish in this trial was to evaluate the potential effects of the dietary Jerusalem artichoke on the health status of the animals. We believe that the parameters we chose to measure were sufficient to evaluate the potential differences that the inclusion of the prebiotic would make.

We would like to point out the fact that the values of Total proteins and Albumin did not present significant differences between treatments. For this reason, we did not get into the details of correlating these results with those of growth performance, for which significant differences were observed.

We did not consider including the values of Globulins and the ratio Albumin/Globulins mainly because we did not assess systemic immunity, but only the mucosal immunity in the intestine by means of counting the amount of goblet cells present. Hence, we would not have had any parameter to correlate these values to in a direct way. 

Point 10: Discussion It needs through revision for more clarity.

Response: Thank you for the suggestion. We have improved the discussion in the revised manuscript in Lines 259-260, 267-272, 275-289, 291-298, 301-303, and 304-306.

Point 11: Conclusion It is not convincing. here write only significant observations that can go to end users. Last two sentence are not needed here better to write what is significant.

Response: The Conclusion has been rewritten.

Lines 330-335: In conclusion, the results of the present study revealed that JA-supplemented diets improved the growth performance, the amount of intestinal mucous cells, the expression of antioxidant-related genes and the disease resistance in juvenile red tilapia against A. veronii. Differences in the effects between the two prebiotic diets were not significant in most of the cases. Therefore, Jerusalem artichoke can be included as a prebiotic in diets for red tilapia.

Reviewer 3 Report

Since fish nutrition is an indispensable element of any production, I consider as a very important any research trying to find new feed additives, or such a research that allows to understand how and whether a given feed additive will bring benefits to production.

The presented research may not be the most innovative or the ones that show a completely new quality, but I believe that it was done correctly, in accordance with "state of art", and the presented results are consistent and quite clearly presented. I read the MS easily and without having to think about each sentence "what the author meant".

Nevertheless, I present a few remarks below:

Title - I understand that the Authors wanted to include as much information as possible in the title, but due to its length it is, in my opinion, far too difficult to understand. I would suggest to rewrite it.

Line 56 – please add „bad” or „not satysfied” before water

Line 91 – What do you mena as a control diet? Please provide any information about the dry feed you youse as a control diet in your experiment.

Line 134 – Gene expression: please provide the information about the samples preservation. Do Authors preserved the samples in liquid nitrogen o rany type of „RNAlater” solution?

Lines 196 – 198 – If something was not statysticaly significant I would suggest not to mention it cause this information will not give a reader any significant meaning.

Line 203 – Could Authors write somewhere in Disscusion i sit better to have higher or lower absorptive area?

Line 227 – But there were no significant differences in the expression of gpx1 between control diet and JA10. Also according to the Figure 2 there were no significant differences between JA5 and JA10 and control diet and JA5 in the case of gr gene. Expresion of cat gene was significantly higher in JA10 according to contro diet and JA5 and there were no significant diferences in expresion of sod gene. Please rewrite the information in results.

Conclusion – I miss a sentence or two aboutwhat Authors would recomend for future experiments with suplementation in red tilapia…to suplement or not suplement? And what about jerusalem artichoke…is there a sens to use i tor not? I Conclusion part you should give a clear statement…do i tor not, or maby U think that more experiments are needed? Plesase suplement the Conclusion with few more sentence.

Author Response

Response to Reviewer 3 Comments

Since fish nutrition is an indispensable element of any production, I consider as a very important any research trying to find new feed additives, or such a research that allows to understand how and whether a given feed additive will bring benefits to production.

The presented research may not be the most innovative or the ones that show a completely new quality, but I believe that it was done correctly, in accordance with "state of art", and the presented results are consistent and quite clearly presented. I read the MS easily and without having to think about each sentence "what the author meant".

Nevertheless, I present a few remarks below:

Point 1: Title - I understand that the Authors wanted to include as much information as possible in the title, but due to its length it is, in my opinion, far too difficult to understand. I would suggest to rewrite it.

Response: Thank you for the valuable comment. We have improved the title after considering reviewer suggestions. The new title is “Effects of Jerusalem Artichoke (Helianthus tuberosus) as a Prebiotic Supplement in the Diet of Red Tilapia (Oreochromis spp.)”.

Point 2: Line 56 – please add „bad” or „not satisfied” before water

Response: Thank you for the suggestion. We have added “poor” in Line 54.

Point 3: Line 91 – What do you mean as a control diet? Please provide any information about the dry feed you use as a control diet in your experiment.

Response: Thank you for your comment. The commercial feed (Hi-grade 9951 Tilapia feed, CPF PCL, Thailand) was used as a control diet. The commercial feed (1 g) was thoroughly mixed with distilled water (1 mL), air dried and stored in 4 °C until further use.

Lines 92-96: The experimental diets (JA5 and JA10) and the control diet were prepared by thoroughly mixing 1 mL of the Jerusalem artichoke (powder dissolved in distilled water) in diets JA5 and JA10 and 1 mL of distilled water with 1 g of feed in the case of the commercial diet (Hi-grade 9951 Tilapia feed, CPF PCL, Thailand), air-dried and stored at 4° C until feeding. ” in the revised manuscript.

Point 4: Line 134 – Gene expression: please provide the information about the samples preservation. Do Authors preserved the samples in liquid nitrogen or any type of „RNAlater” solution?

Response: Thank you for your comment. The samples were collected, quick-frozen on ice, and stored at -80 °C until RNA isolation.

Point 5: Lines 196 – 198 – If something was not statistically significant I would suggest not to mention it cause this information will not give a reader any significant meaning.

Response: Thank you for the comment. We decided to add these results in text despite the lack of significance because we found the numerical trends interesting.

Point 6: Line 203 – Could Authors write somewhere in Discussion is it better to have higher or lower absorptive area?

Response: Thank you for the comment. The discussion part on the results of the intestinal histology has been extended.

Lines 294-298: In terms of intestinal morphology, prebiotics may increase the absorptive area of the gastrointestinal tract based on changes in the microvillus height, among others [51–53]. Higher absorptive surface areas and higher microvilli densities in the intestine result in an increase in the nutrient absorption [54], which may result in increased weight gain and feed efficiency [55].

Lines 301-302: which was also observed in red drum fed dietary prebiotics during 8 weeks [53].

Lines 304-306: Hence, a longer experimental feeding period would be desired in order to assess the intestinal morphology.

Point 7: Line 227 – But there were no significant differences in the expression of gpx1 between control diet and JA10. Also according to the Figure 2 there were no significant differences between JA5 and JA10 and control diet and JA5 in the case of gr gene. Expresion of cat gene was significantly higher in JA10 according to contro diet and JA5 and there were no significant diferences in expresion of sod gene. Please rewrite the information in results.

Response: Thank you for the suggestions. The results section ‘Antioxidant genes expression’ has been rewritten.

Lines 234-239: The prebiotic diet JA5 caused a significant increase (p < 0.05) in the expression of gene gpx1 (2.69-fold and 1.36-fold) compared to the Control and JA10 diets, while diet JA10 resulted in a significant increase in the expression of genes gst (4.2-fold) and gr (3.17-fold) compared to the Control diet. The expression of cat in fish fed the prebiotic diet JA10 was significantly higher (p < 0.05) (1.96-fold) than in that of fish from the control diet, while the expression of sod did not present significant differences between groups (p > 0.05).

Point 8: Conclusion – I miss a sentence or two about what Authors would recommend for future experiments with supplementation in red tilapia…to supplement or not supplement? And what about jerusalem artichoke…is there a sense to use it or not? I Conclusion part you should give a clear statement…do it or not, or maybe you think that more experiments are needed? Please supplement the Conclusion with few more sentence.

Response: Thank you for the suggestions. The Conclusion has been rewritten.

Lines 330-335: In conclusion, the results of the present study revealed that JA-supplemented diets improved the growth performance, the amount of intestinal mucous cells, the expression of antioxidant-related genes and the disease resistance in juvenile red tilapia against A. veronii. Differences in the effects between the two prebiotic diets were not significant in most of the cases. Therefore, Jerusalem artichoke can be included as a prebiotic in diets for red tilapia.

Round 2

Reviewer 2 Report

author has addressed all comments and now it is in good shape to publish.